# A Simple Approach to Estimate the Drag Coefficients of a Submerged Floater

**DOI:** 10.3390/s23031394

**Published:** 2023-01-26

**Authors:** Yuval Hoffman, Liav Nagar, Ilan Shachar, Roee Diamant

**Affiliations:** Hatter Department of Marine Technologies, University of Haifa, Haifa 3498838, Israel

**Keywords:** drag force, hydrostatic force, thruster force, submerged floaters

## Abstract

The calculation of the drag force is a fundamental requirement in the design of any submerged system intended for marine exploration. The calculation can be performed by analytic analysis, numerical modeling, or by a direct calculation performed in a designated testing facility. However, for complex structures and especially those with a non-rigid design, the analytic and numerical analyses are not sufficiently accurate, while the direct calculation is a costly operation. In this paper, we propose a simple approach for how to calculate the drag coefficient in-situ. Aimed specifically at the complex case of elastic objects whose modeling via Computer-Aided Design (CAD) is challenging, our approach evaluates the relation between the object’s speed at steady-state and its mass to extract the drag coefficient in any desired direction, the hydro-static force, and, when relevant, also the thruster’s force. We demonstrate our approach for the special case of a highly complex elastic-shaped floater that profiles the water column. The analysis of two such floaters in two different sea environments shows accurate evaluation results and supports our claim for robustness. In particular, the simplicity of the approach makes it appealing for any arbitrary shaped object.

## 1. Introduction

Floaters are a valuable tool for probing the water column. They can provide water current estimation [1] by tracking their drift motion over time, characterize internal waves [2] by observing spatial changes in the depth of the bathymetrical layer, or monitor changes in the marine environment by measuring temperature profiles over time with fine resolution [3]. Examples of floaters include profiling floats such as the Argo floats [4], which traverse the water column from the seabed to the sea surface for conductivity–temperature–depth (CTD) measurements, and Lagrangian floats, which are designed to drift with the water current [5]. For both types, calculating the drag coefficient is important information for improving the system design. Floaters are also used to evaluate the water current’s properties [6]. Here, the analysis of the floater’s drag is essential to filter out friction forces from the measurements. The drag is an attribute of how well the float drifts in the water current and is a function of the float’s shape and size. A perfect, rounded, neutrally buoyant float with a narrow edge on its endfire will sense little resistance as it traverses through the water, while a square float with a large surface facing the drifting direction will sense high friction-like resistance.

The calculation of the drag coefficient of a submerged object is performed either by analytical modeling, using numerical finite-elements simulations, or directly in designated testbeds. The first analytical expression to calculate the drag coefficient is the seminal work by Jean le Rond d’Alembert in 1752, who calculated the drag force of a flow acting on a cylinder shape object using the potential flow theory (incompressible and inviscid, for a Reynolds number Re >>1). The analysis yielded a zero drag force, which was in contrast to the experiments made. This mismatch is referred to as the *D’Alembert’s paradox* and is due to neglecting the water boundary layers. The later theoretical work of Munk and Glauert in the 1920s [7] presented the base for *thin airfoil theory*, which uses potential flow theory to calculate the drag and lift force on a thin wind. For Re <<1, for which the inertia at the N–S equation can be neglected, the so-called *stock flow* or the *creeping flow* linear equation is obtained, which can be solved in a close form to yield the drag coefficient [8]. That being said, analytics methods are limited to simple geometries and flow regimes, and current methodology leans towards numerical evaluation.

In the practical case where the object studied is of complex geometry or when the object travels through a complex flow regime, a common technique to determine the drag coefficients is the Computational-Fluid Dynamics (CFD) method [9]. The framework of CFD allows the estimation of the drag coefficients directly from its 3D model and thus allows performance evaluation even before its manufacturing to reduce the price of design. The method fragments the fluid volume into discrete cells, where each cell equation (i.e., Navier–Stokes, Euler equations) is solved iteratively while respecting the overall boundary condition. As such, a CFD simulation takes a long time to converge and requires advanced computing equipment. Moreover, for objects of non-rigid structures, such as a floater with many sharp edges and a flexible surface design, the model of the object may be too simplified, and the numerical calculation may lead to a significant error. In such cases, designers turn to direct drag measurements [10].

Measurements of drag coefficients are mostly performed in a tow tank (cf. [10]) to test the drag over the object itself or its smaller scale model. This kind of testing facility involves a Planar Motion Mechanism (PMM) that is stationed above the tank. This is an electromechanical device used for maneuvering studies, which moves in a specific regime while directly measuring the force induced on the object tested. While results are accurate, the process requires a large and expensive facility that is only profitable for shipyards or large companies. An alternative approach involves the use of a reduced-order model (ROM). Proposed by both Morrison and Younger [11] and Cely et al. [12], the hydrodynamic coefficient is calculated using a simple experiment. The tested object, in this case a remotely operated vehicle, is mounted on springs inside a water tank to measure its free decay. From the free decay measurements, the damping coefficients is extracted using non-linear least squared (NL-LS) methods. A similar approach by Eng et al. [13] uses a pendulum ROM to calculate the hydrodynamic coefficient. Still, the ROM offers only an approximation. To solve this, alternative approaches are offered to experimentally calculate the drag coefficient.

The use of experiments offers the possibility to test the object in realistic conditions. Caccia et al. [14] demonstrated the calculation of the drag of a thruster-aided vehicle in a sea experiment. The method involves measuring both the depth of the vehicle and its thruster’s power. Then, by measuring the voltage–thrust relation in a complementary experiment, the method accounts for the propeller–hull and propeller–propeller effects that are usually neglected in standard numerical models. Still, a hard assumption is made claiming the drag force is omnidirectional, even though it may hold strong directional dependencies [15]. For example, this assumption may break when the object includes moving or flexible parts or has a non-uniform section area body. As such, the above approach is limited to a rigid body with a uniform section area.

In this work, we offer a low-complexity approach for the in-situ measurement of the drag coefficient. Our method defines a calibration test for a tested vehicle to measure the relation between its motion and its weight. Since the steady state velocity of the vehicle is a function of its drag, we evaluate the parameters of the speed–weight relation to extract the drag force and the hydrostatic force. To get a good estimation of the former relation, the calibration test should include traveling at different speeds. We solve this by performing multiple runs with different weights on the vehicle. We showcase our approach on a self-made floater that uses a thruster to ascend and a parachute to slow down its descending speed, thus decreasing the floater’s energy consumption. Being long and narrow, the drifter’s surface-to-weight ratio reduces the effect of the environment. To examine this, we tested the operation of the floater in two different sea experiments and analyzed its ascending and descending motion to yield a calculation of the drag. The results show a match between the relation assumed in our analysis between the squared speed and the floater’s mass and a fair accuracy when comparing the estimated values for the hydrostatic force and thruster force to their direct measurement. Our novelty is in the concept of using weights to yield a different ascending/descending speed for the drifter without changing the other parameters of the system. In this way, we could harness the relation between the speed and drag. We argue that, without using weights, to evaluate the drag, one should control the system actively, through e.g., an active control mechanism, which would complicate the trial greatly up to the point that it could not be performed in-situ.

The structure of this paper is as follows. A literature review from Instrumentation and Measurement journals is offered in Section 2. In Section 3, we introduce our system model, main assumptions, and the structure of our floater as a test case. In Section 4, we describe our in-situ approach for measuring the drag coefficient. Results from two sea experiments are presented in Section 5, and conclusions are drawn in Section 6.

## 2. Survey of Instrumentation and Measurements Techniques

Considerations of how to measure the drag force over a tested system in a wind tunnel are presented in [16], where the dimensions of the measuring unit are emphasized. Aiming for a small testing unit, a suspension force-measuring system (SFMS) is proposed in [17]. Results show a sensitivity of 0.25 mV/N for drag force measurements, but a long calibration process is required. The measuring of the drag force is similar in essence to that of the friction force. Techniques for accurately measuring the friction coefficient are surveyed in [18,19]. For example, a common standard in this field is described in [20], where acoustic, optical, or tread sensors are installed in the runway to measure friction-related parameters. The existing methods require testing facilities, and gaps are identified regarding how to consider environmental conditions in the measurement. This is specifically important when operating in a sea environment.

The measurement of the drag coefficient for marine applications has been the focus of many instrumentation research works. In [21], the turbulence statistical method has been adopted to measure the drag force of a piezoelectric-based device for the wall friction drag reduction of micro underwater vehicles. The tests include a source for turbulence burst, a current meter, and the placement of the tested device downstream at a target Reynolds number. Particle image velocimetry is used to visualize the fluid and measure the turbulence. The impact of the drag can also be observed from the relation between the speed of the tested vehicle and the feedback received from its motor, namely the motor’s current and the propeller’s speed. This is similar to the method in [22] that maps between this relation based on fuzzy logic and an adaptive filter. Another form of impact of the drag is investigated in [23] to design a non-linear dynamic controller for AUVs. The review in [24] describes the different considerations in designing an underwater vehicle. For example, in [25], a design of a glider designed to glide smoothly in the water is proposed, and a controller that changes the vehicle’s angle of attack to reduce the drag, for example, is proposed. Similarly, design concepts based on drag for AUVs are described in [26].

A few techniques exist for measuring the drag and its impact on underwater vehicles. In [27], the effect of the drag is measured in a pool experiment in a lab for an underwater fan-wing thruster for both towing and self-driving modes. The vehicle velocity and the vertical source are analyzed for the latter, whereas a special device for measuring the drag force is used for the former. Results are found to be in agreement with simulation analysis. In [28], experiments measure the hydrodynamic forces on an AUV as a function of its acceleration and velocity. Static measurement in a straight line and dynamic tests involving pitch changes are compared to measurements in a water tunnel. However, the method requires stability and thus does not apply in realistic sea conditions except for deep sea tests. To prove the design of an AUV with soft legs and a soft inflatable morphing body for position keeping, ref. [29] measured the drag and lift of a vehicle in a flow channel using an actuator and a servo motor as well as laser Doppler velocimetry for velocity measurement. Operating at sea, a horizontal drag measurement platform was used in [30] for turbulence observation with the help of an extremely stable platform. A method to evaluate the active and passive drag of a swimmer is proposed in [31]. The method relies on residual thrust measurements, i.e., the difference between the propulsive and resistive forces, performed in a water flume that allows the flow velocity to be adjusted along with an evaluation of the relation between the swimmer’s propulsion and drag. In [32], the Levenberg–Marquardt algorithm is used to estimate the drag coefficient of an AUV in a calibrated pitot tube. While accurate measurements are obtained, to the best of our knowledge, there is no robust method for estimating the drag of any object in-situ in realistic sea conditions.

## 3. System Model

Our system includes a complex-shaped floater that profiles the water column using a thruster while collecting acoustic and CTD measurements. The floater includes means to measure its instantaneous depth and is time-synchronized prior to deployment. The calibration of the floater’s drag involves a few ascend/descend cycles before changing the floater’s weights. Then, the floater’s depth profile is analyzed offline.

As illustrated in Figure 1, the floater considered is a 3-inch cylinder containing electronic components: a hydrophone; pressure, salinity, and temperature sensors; and a thruster. The floater’s depth profiling is performed by defining a lower and upper limit to the floater. Specifically, being negatively buoyant, the floater operates its thruster to ascend, and stops the thruster to descend. Then, upon reaching a lower depth limit, the floater operates its thruster and ascends, and upon reaching its upper depth limit, the floater stops the thruster and descends. To reduce battery usage, the floater includes a parachute-like tarpaulin sheet that, much like an umbrella, opens when the floater sinks and closes when it ascends. This is made possible by three rigid elements given the fabric flexibility against the water pressure. As such, the floater holds its complex shape, and its drag coefficient is hard to calculate numerically. In particular, its flexibility makes the floater’s boundary conditions complex to evaluate. A picture of the floater during operation is shown in Figure 2.

## 4. Our Drag Coefficient Measuring Technique

The key idea behind our method is the utilization of the relation between the floater’s steady-state speed and its weight. We perform the drag estimate in steady state to eliminate the drag coefficient dependency in the system’s acceleration. Since there is a square relation between the drag force and the vehicle’s velocity, the ratio between the vehicle’s speed and its weight should be also a square relation. Then, by quantifying the parameters of the latter relation, we can evaluate the drag force as well as the floater’s hydrostatic force. The specific test thus includes allowing the floater to rise and submerge while collecting depth profiles to evaluate the floater’s speed and repeating this process multiple times with different weights to yield several points for an offline speed–weight ratio evaluation.

### 4.1. Method Formalization

Let $\rho_{p}$ be the plumbum density, $v_{p}$ be the volume of the floater’s weights, $\rho_{w}$ be the water density, and $v_{b}$ be the volume of the floater. Also let, $C_{\mathrm{up}}$ be the quadratic drag coefficient, $A_{\mathrm{up}}$ be the floater’s section area while rising, and $F_{\mathrm{thruster}}$ be the force obtained from the thruster. The force balance on the floater while rising is
(1)$$\begin{array}{ccc} F_{x} & = & \left(m+\rho_{p}v_{p}\right)\ddot{x}=-mg+\rho_{w}v_{b}g˘\left(\rho_{p}-\rho_{w}\right)v_{p}g\\ & - & \frac{1}{2}\rho_{w}C_{\mathrm{up}}A_{\mathrm{up}}{\dot{x}}^{2}+F_{\mathrm{thruster}}\;, \end{array}$$
where *m* is the floater’s mass.

#### 4.1.1. Steady State: Ascending

To formalize the steady-state behavior, the following definitions are required. The quadratic damping coefficient is
(2)$$\Psi_{\mathrm{up}}=\frac{1}{2}\rho_{w}C_{\mathrm{up}}A_{\mathrm{up}}\;.$$
The total hydrostatic force on the floater body is
(3)$$\Lambda =-mg+\rho_{w}v_{b}g\;.$$
The total hydrostatic force acting on the sinkers is
(4)$$\kappa =\left(\rho_{p}-\rho_{w}\right)v_{p}g\;.$$
Then, we rewrite (Equation 1) as
(5)$$\left(m+\rho_{p}v_{p}\right)\ddot{x}=\Lambda -\kappa -\Psi_{\mathrm{up}}{\dot{x}}^{2}+F_{\mathrm{thruster}}\;.$$
At steady state, the velocity is
(6)$${\dot{x}}^{2}=-\frac{\kappa }{\Psi_{\mathrm{up}}}+\frac{\Lambda +F_{\mathrm{thruster}}}{\Psi_{\mathrm{up}}}\;.$$
Note that in our calibration experiments, the right-most term at the right hand side of (Equation 6) is constant, while the other terms are variable.

#### 4.1.2. Steady-State: Descending

When the floater submerges, the drag force changes its direction, and the floater’s force, $F_{\mathrm{thruster}}$, vanishes. Then,
(7)$$\begin{array}{ccc} F_{x} & = & \left(m+\rho_{p}v_{p}\right)\ddot{x}=-mg+\rho_{w}v_{b}g\\ & - & \left(\rho_{p}-\rho_{w}\right)v_{p}g+\frac{1}{2}\rho_{w}C_{\mathrm{down}}A_{\mathrm{down}}{\dot{x}}^{2}\;, \end{array}$$
where $C_{\mathrm{down}}$ and $A_{\mathrm{down}}$ are the quadratic drag coefficient and the floater’s section area while descending, respectively. The damping coefficient can then be written as
(8)$$\begin{array}{c} \Psi_{\mathrm{down}}=\frac{1}{2}\rho_{w}C_{\mathrm{down}}A_{\mathrm{down}}\;. \end{array}$$
We rewrite (Equation 7) as
(9)$$\begin{array}{c} \left(m+\rho_{p}v_{p}\right)\ddot{x}=\Lambda -\kappa +\Psi_{\mathrm{down}}{\dot{x}}^{2}\;, \end{array}$$
and the descending steady state becomes
(10)$$\begin{array}{c} {\dot{x}}^{2}=\frac{\kappa }{\Psi_{\mathrm{down}}}-\frac{\Lambda }{\Psi_{\mathrm{down}}}\;. \end{array}$$

The above analysis shows a linear relation between ${\dot{x}}^{2}$ and κ. This linear relation holds with small Reynolds numbers, as in the case of the drifter. We use this observation as a validation metric in the following.

## 5. Experimental Evaluation

### 5.1. Experimental Setup

To demonstrate the applicability of our approach in realistic scenarios, we performed two sea experiments using two different floaters. The first experiment was conducted in the Mediterranean Sea in September 2021 across the shores of Haifa at a shallow water environment of depth 10 m. The sea state was 2, and the water current was roughly 2 knots. The floater was deployed from a small vessel and performed ascend/descend profiles at the depth range of 1 m and 7 m. After five profiles, the floater automatically ascended to the surface where additional plumbum weights were added. We note that, due to their dense mass with respect to the magnitude of the water current, the drag of the weights can be neglected. The process then repeated itself for two more runs of five profiles each. In Table 1, we give the mass of the added weights and the measured in-water mass for this experiment. During the entire operation, the floater was observed from the distance by a snorkeler who made sure that, during its profiles, the floater did not touch the seabed or reach the surface. An example of the depth profile as collected by the floater is shown in Figure 3.

The second experiment took place in the Red Sea, Eilat, Israel in January 2022. This experiment was conducted in deeper water at 30 m depth. A different floater than the one used in the first experiment but with a similar shape was handled by scuba divers who descended to 20 m depth and operated the floater. The floater continuously performed profiles between 5 m depth and 20 m, and every five ascend–descend profiles, the divers changed the floater’s weights. The use of different weights between the two experiments was to compensate for the different salinities of the two seas explored, as well as to add diversity to the results. We note that the drifter reached a steady state speed after less than 1 m from the maximum point of rising or descending, and thus testing at lower depth had no influence on performance except for additional data points for better speed estimation via regression. In total, five weight changes were made, and the experiment lasted for 50 min. The mass of the added weights and the measured in-water mass for this experiment are given in Table 2. An example of the depth profile as collected by the floater is shown in Figure 4.

The outcome of the two experiments was a measure of the relation between the floaters’ measured velocity at steady state, $\dot{x}$, and the measured mass of their plumbum weights, κ. To that end, the floaters’ initial buoyancy, i.e., before adding weights, was made negative, causing the floater to sink with no thruster force—that is, Λ−κ<0. The offline analysis included averaging the five ascend–descend profiles to yield the average steady-state ascend and descend speed.

### 5.2. Experimental Results

By (Equation 6) and (Equation 10), the relation between the velocity and the weights’ mass is quadratic. To parametrize this relation and to extract the drag force, we performed a linear regression over the measured square velocity and the weights’ mass. The results of this linear fitting for the Haifa trial, along with the averaged measurements, are shown in Figure 5a. We observe a good fit between the measured relation and the linear fitting from which the following parameters are drawn: $\Psi_{\mathrm{up}}=23.9$ kg/m, $\Psi_{\mathrm{down}}=188.5$ kg/m, Λ=−0.68 N, $F_{\mathrm{thruster}}=7.07$ N. The results for the Eilat trial are given in Figure 5b. These results also show a good fit for the linearization attempt, and the following parameters are drawn: $\Psi_{\mathrm{up}}=16.8$ kg/m, $\Psi_{\mathrm{down}}=301.2$ kg/m, Λ=−1.11 N, $F_{\mathrm{thruster}}=7.4$ N.

We note the large differences in the estimated drag force for the descending and ascending directions caused by the operation of the parachute. Comparing the results between the two experiments, we observe a change in the measured drag force (ascending and descending). This difference is due to a different parachute size (for descending) and a different thruster (for ascending).

### 5.3. Validation of Results

The small linearization error we observe in both sea experiments confirms our above analysis. Still, there is a need to validate the obtained results. Unfortunately, as noted in Section 1, directly measuring the drag coefficient of the two floaters poses a significant challenge since the flexibility of the parachute requires a steady-state measurement in both ascending and descending directions, which is hard to achieve in a testing facility. However, it is relatively straightforward to obtain a direct measurement of the hydrostatic force and the floater’s thruster’s force. Thus, we validate our results by comparing the direct and indirect measurement of these two latter characteristics.

The hydrostatic force can be measured by observing which is the minimum force required to make the floater sink. We provide a bound for this force by adding small weights to the floater as listed in Table 1 and Table 2 until the floater sinks. In particular, we deployed the two floaters in a 9 m × 3 m salt water pool of 3 m depth and balanced the hydrostatic force to be slightly positive. After fine tuning, we observed that, when adding a weight of 129 grams to the floater used in the first experiment and a weight of 210 gram to the floater used in the second experiment, the hydrostatic force of the two floaters becomes negative and the floaters sink. As a result, the hydrostatic force can be bounded within the range −1.15 N ≤Λ<0 for the floater used in the first experiment and within −1.877 N ≤Λ<0 for the floater used in the second experiment. We note that a value in-between these ranges was obtained for both floaters in the in-situ measurements.

We note that a similar operation for measuring the thruster’s force is harder since this value depends on the floater’s speed. Still, assuming the speed difference along the floater’s depth profile is negligible, we can compare the estimated thruster’s force, $F_{\mathrm{thruster}}$, to its static measurement. To that end, we balanced the floater at a slight positive hydrostatic force and measured the above-water height of the floater. We then operated the thruster—an operation that pushed the floater up and increased its above-water height. Comparing the above-water heights at the two operation modes, the thruster’s force is evaluated by
(11)$${\hat{F}}_{\mathrm{thruster}}=\Delta h\pi \frac{D^{4}}{4}\rho_{w}g\;,$$
where Δh is the height difference between the thruster-on and the thruster-off modes, and *D* is the floater’s diameter. For the floater used in the first experiment, we obtained Δh=0.1 m, which translates to ${\hat{F}}_{\mathrm{thruster}}=8.6$ N (vs. 7.07 N in the in-situ measurement). For the floater used in the second experiment, we obtained Δh=0.13 m, which translates to ${\hat{F}}_{\mathrm{thruster}}=11.2$ N (vs. 7.4 N in the in-situ measurement). While in both cases, we argue that the results are very close, it is apparent that the analysis for the Haifa floater achieved better accuracy. This is because the floater used in the second experiment was equipped with a faster, and thus lower efficiency, thruster, which somewhat goes against our above assumption that the dynamic force and the static one are similar.

### 5.4. Discussion

In this work, we have demonstrated an in-situ approach for measuring the drag coefficient in two sea environments and for two different floaters. The good results obtained in both cases support our claim for robustness and show that our approach can fit different depth profiles and can be implemented over different platforms. The difference between the values obtained for the two floaters, despite their seemingly similar shapes, emphasizes the need for an in-situ measurement such as ours rather than settling for numerical evaluation, especially when analyzing elastic shapes.

We admit that our method is limited by a few factors. First, the measurement of the object’s speed should be performed in steady state, which requires careful planning of the calibration test. Second, the method is sensitive to strong water currents or waves that may change the structure of the examined object. For example, a strong water current may partially fold our floaters’ parachute. Last, our assumption that the thruster’s force in a dynamic scenario is similar to that in a static scenario only holds when the thruster operates at slow speed. As a result, the test for the drag coefficient must be planned as a calibration test rather than performed as part of an ongoing operation. To handle such discrepancies, future work may account for the system efficiency in the calculation of the speed–mass relation and for cases where the drag coefficient is not constant for different speeds.

## 6. Conclusions

In this work, we outline the details of an in-situ approach for calculating the drag force of an arbitrary shaped object. Our approach utilizes the expected quadratic relation between the object’s speed and its mass. With no modeling of the object tested, and using a simple setup to evaluate the parameters of this relation, our method can accurately measure the object’s drag force in any direction, its hydrostatic force, and the thruster’s force if a thruster is included. Results from two sea experiments using thruster-operated floaters with an elastic shape demonstrated the applicability of our approach and revealed a good match between our in-situ evaluation and a direct measurement of the hydrostatic force and the thruster’s force. Our solution is suitable to systems with a small Reynolds number and at steady state. Future work would explore how to extract the drag coefficient also in the non-linear regime.

## Figures and Tables

**Figure 1 sensors-23-01394-f001:**
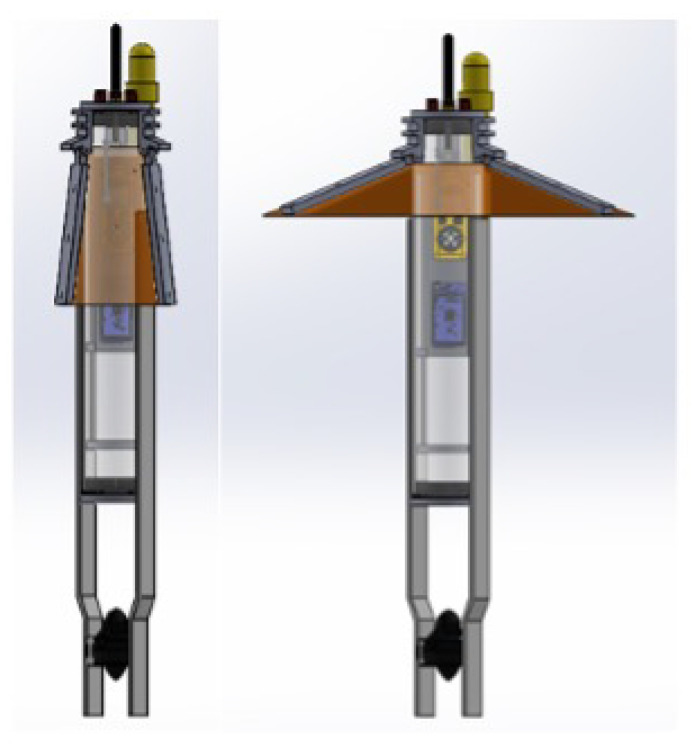
A model of the considered floater. In the left panel, the floater’s arms are close, while in the right one, they are open.

**Figure 2 sensors-23-01394-f002:**
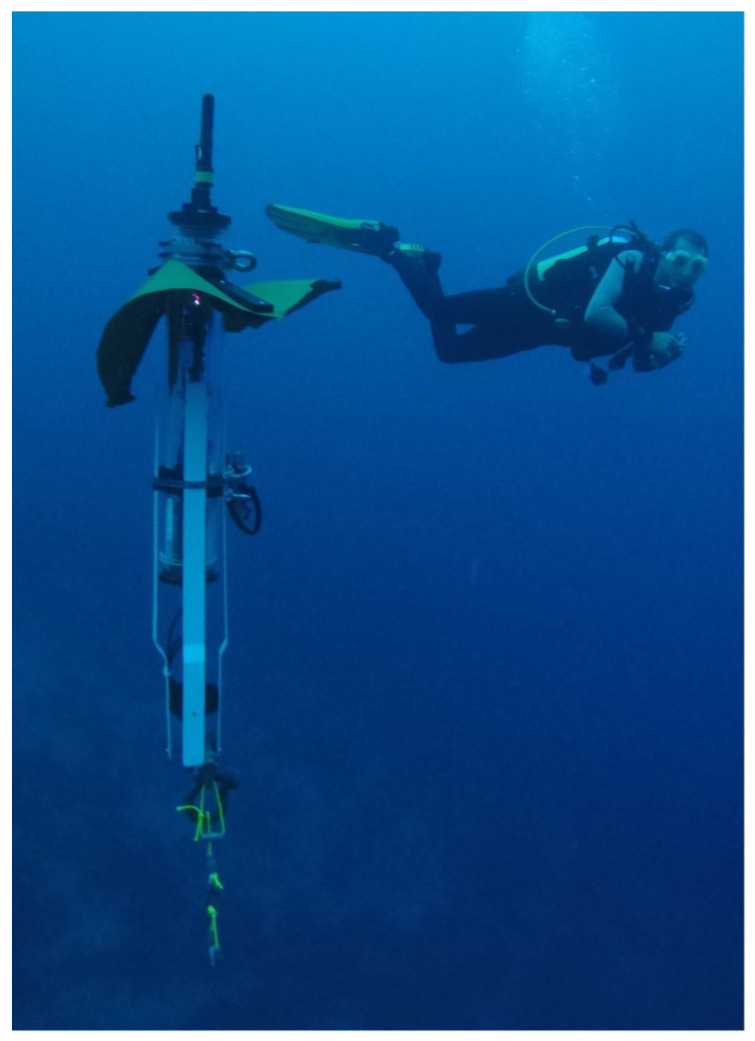
A picture of the floater in operation with a scuba diver in the background for scale. In this picture, the parachute-like tarpaulin sheet is in a semi-open stage.

**Figure 3 sensors-23-01394-f003:**
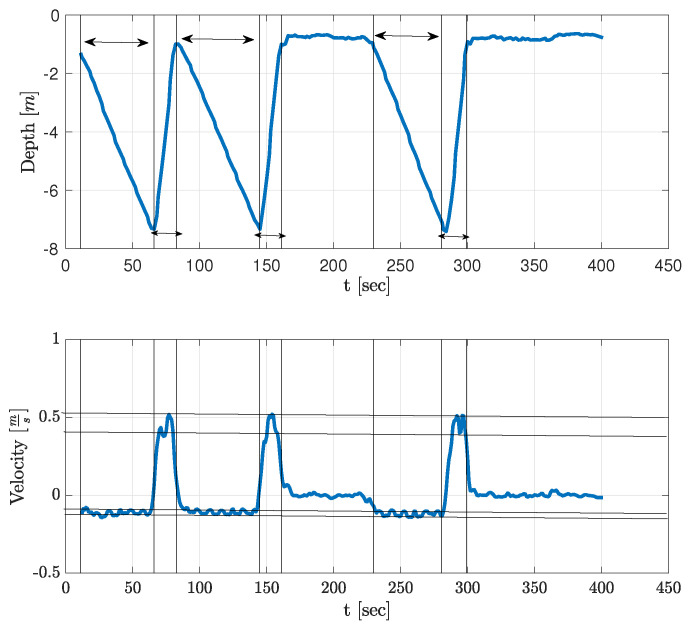
Example of three depth profiles recorded by the floater during the Haifa sea experiment. Arrows mark rising and descending periods.

**Figure 4 sensors-23-01394-f004:**
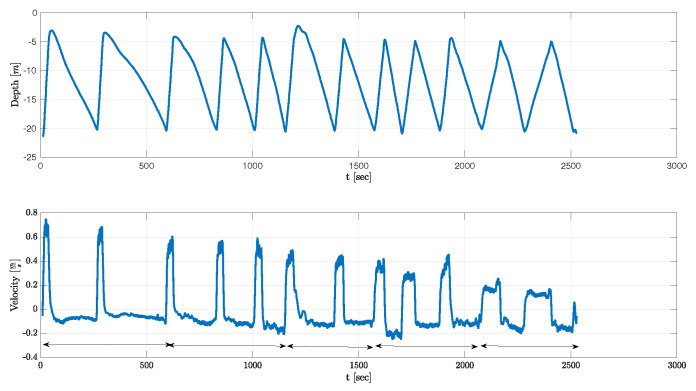
Example of multiple depth profiles recorded by the floater during the Eilat sea experiment. Arrows mark periods for each set of weights.

**Figure 5 sensors-23-01394-f005:**
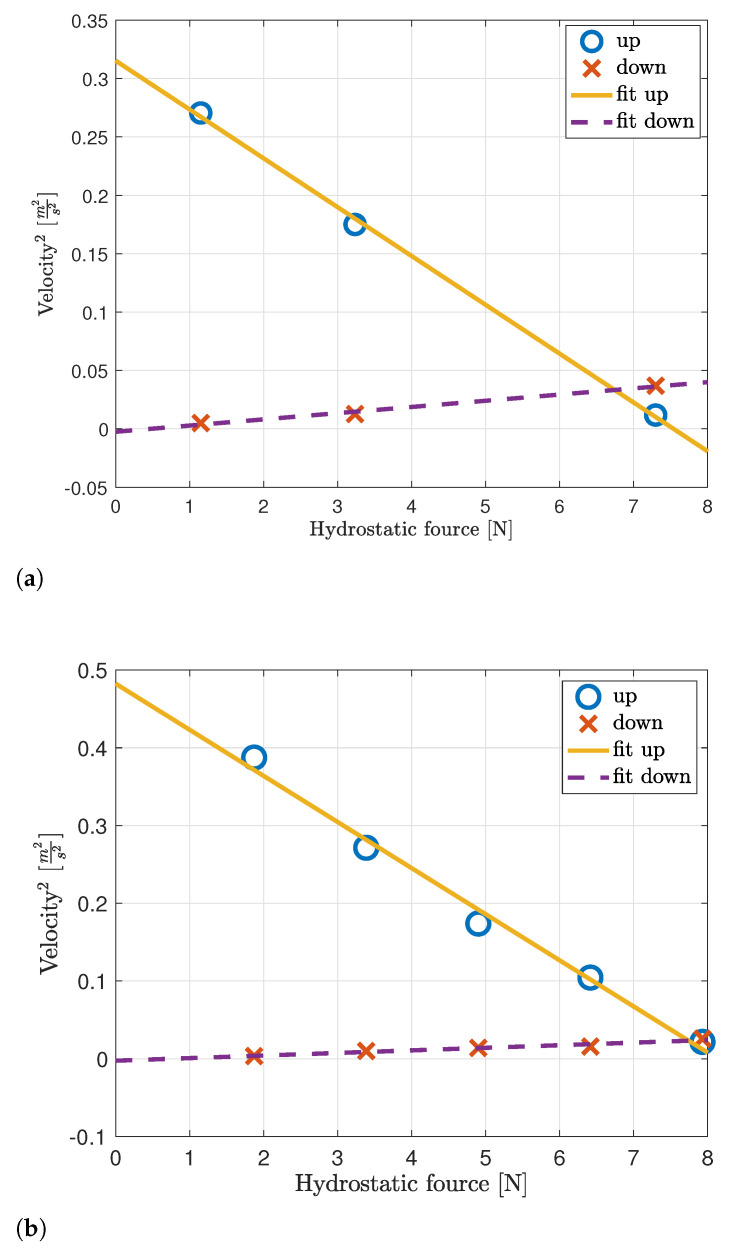
Squared floater’s velocity vs. weights’ mass: measured and linear fit. Results are shown separately for ascending and descending. (**a**) Haifa, (**b**) Eilat.

**Table 1 sensors-23-01394-t001:** Weights’ mass and underwater mass during the Haifa experiment.

Profile Number	Weights’ Mass [g]	Underwater Weight [N]
1	129	1.15
2	363	3.23
3	816	7.3

**Table 2 sensors-23-01394-t002:** Weights’ mass and underwater mass during the Eilat experiment.

Profile Number	Weights’ Mass [g]	Underwater Weight [N]
1	210	1.877
2	380	3.396
3	550	4.915
4	720	6.435
5	890	7.954

## Data Availability

Not applicable.

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
