# Peer review of "A Simple Approach to Estimate the Drag Coefficients of a Submerged Floater"

_sensors, 2023, doi:10.3390/s23031394_

Round 1
Reviewer 1 Report
1. in line 74 -76, in this work, we offer a low complexity approach for in-situ measurement of the drag coefficient. Our method defines a calibration test for a tested vehicle to measure the relation between its motion in steady-state and its weight. The environment is dynamic and diversity changed, how to except these factors from different motion speed.
2. In context, Since there is a square relation between drag force and the vehicle’s velocity, the ratio between the vehicle’s speed and its weight should be also a square relation. The former half is obvious, but the late half vehicle's speed and weight relationship is so precise.
3. Drag is some also related with acceration . It was some curiously that if speed up or slow down that will affect the result of equation(11), the thruster’s force relation.
4. Since the result is some kind of linear related , if there are some limit of speed scope, that will be thought.
5. In the conclusion, it maybe more discussed about the merit and limitation of this simplified method.
Author Response
Please see our one-to-one response below:
1. in line 74 -76, in this work, we offer a low complexity approach for in-situ measurement of the drag coefficient. Our method defines a calibration test for a tested vehicle to measure the relation between its motion in steady-state and its weight. The environment is dynamic and diversity changed, how to except these factors from different motion speed.
Answer: We agree with the reviewer that the marine environment changes and may effect differently the drifter. However, being of long and narrow shape, the drifter’s surface to weight ratio highly reduces the effect of the environment. We see an evidence to that by obtaining similar results when testing in two different sea environments: the Mediterranean Sea and the Red Sea, both of different salinity and temperature. We have clarified this in the revised text.
2. In context, Since there is a square relation between drag force and the vehicle’s velocity, the ratio between the vehicle’s speed and its weight should be also a square relation. The former half is obvious, but the late half vehicle's speed and weight relationship is so precise.
Answer: We understand the question is regarding the novelty of the work. The use of weights is set to yield different ascending/descending speed for the drifter without changing the other parameters of the system. This way, we could harness the relation between the speed and drag. We argue that, without using of weights, to evaluate the drag one should control the system actively through e.g., an active control mechanism, which would complicate the trial greatly up to a point that it could not be performed in-situ. We have clarified this in the revised text.
3. Drag is some also related with acceleration. It was some curiously that if speed up or slow down that will affect the result of equation(11), the thruster’s force relation.
Answer: The reviewer is correct: the drag coefficient does depend on the acceleration of the system. For this reason, we filtered out from the experiment only results at steady state where the speed and the drag coefficient are constant. We have clarified this in the revised text.
4. Since the result is some kind of linear related , if there are some limit of speed scope, that will be thought.
Answer: Please note that in small Reynolds numbers, the drag coefficient does not depend on the system’s speed, but such dependency occurs in high Reynolds numbers. In our case, the Reynolds number is small. We have clarified this in the revised text.
5. In the conclusion, it maybe more discussed about the merit and limitation of this simplified method.
Answer: Based on the reviewer’s request, we have extended our conclusion with more insights on the performance of the system.
Reviewer 2 Report
Dear Author(s),
Your manuscript entitled 'A simple approach to Estimate the Drag Coefficients of a submerged floater' has drawn certain conclusions with some limitations. It would be more informative as it is an experimental finding. Find a few comments below:
1. Why author has used different weight masses for two different experiments?
2. Line: 216-219. “Comparing the results between the two experiments, we observe that while the drag force is fairly similar between the floaters when ascending, it is very different when descending. We argue that the difference between the two experiments in the ascending direction is due to the use of different thrusters, each of which induces a different thruster’ force,”
i. Contradictory statement found. Please check and do the necessary.
ii. If thrusters were the same for both of the experiments, then whether drag force for both would be the same or different?
3. From the study, it has been learned that this study is limited up to steady-state condition. Then, author should add the future scope as well for researchers in both the abstract and conclusion sections.
Author Response
Please see our one-to-one response below:
1. Why author has used different weight masses for two different experiments?
Answer: We see the reviewer’s point. The use of different weights was to compensate for the different salinity of the two seas explored, as well as to add diversity to the results. We have clarified this in the revised text.
2. Line: 216-219. “Comparing the results between the two experiments, we observe that while the drag force is fairly similar between the floaters when ascending, it is very different when descending. We argue that the difference between the two experiments in the ascending direction is due to the use of different thrusters, each of which induces a different thruster’ force,”
i. Contradictory statement found. Please check and do the necessary.
Answer: We thank the reviewer for his close observation. We have changes the text accordingly. The revised text reads: “Comparing the results between the two experiments, we observe a change in the measured drag force (ascending and descending). This difference is due to a different parachute size (for descending), and a different truster (for ascending).”
ii. If thrusters were the same for both of the experiments, then whether drag force for both would be the same or different?
Answer: Indeed, we argue that the difference in the ascending drag force is only due to the different thrusters used. That is assuming similar water densities of course.
3. From the study, it has been learned that this study is limited up to steady-state condition. Then, author should add the future scope as well for researchers in both the abstract and conclusion sections.
Answer: We agree with the reviewer. Our work is limited to the steady state condition as well as for small Reynolds numbers where the drag coefficient does not depend on the system’s speed. We have clarified this in the revised text, and have added future work in the conclusion.
Round 2
Reviewer 2 Report
Dear Authors,
Thank you for attending to all queries.
Regards.